# One-Shot Federated Aggregation of Generalized Embeddings for Edge Environments

## Abstract

Federated learning (FL) enables collaborative training in decentralized data; however, traditional multi-round protocols incur high communication costs. One-shot FL (OFL) reduces communication cost by limiting interaction to a single round. Existing OFL methods often suffer from poor accuracy and heavy client-side computation, making them unsuitable for resource-constrained edge devices. This paper introduces a novel OFL framework, FedAGE (Federated Aggregation of Generalized Embeddings), in which clients transmit latent representations derived from a shared frozen encoder and a portion of the model instead of full weights. This design offloads expensive computation to the server, drastically reducing the client's overhead of computation, while retaining essential discriminative features within the shared embeddings. FedAGE transfers knowledge to the server through a progressive distillation framework, incorporating weighted soft labels, ensemble distillation, and knowledge mixing to mitigate catastrophic forgetting. Extensive experiments are carried out on the five benchmark datasets. The proposed FedAGE consistently outperforms OFL baselines, achieving up to 46.4% higher accuracy, under high heterogeneous partitioning. The experiments evident that FedAGE's performance is consistent across various levels of data heterogeneity. Also, our analysis shows at least a 66% reduction in client-side computational overhead, measured in GigaFLOPs. These findings confirm FedAGE as a viable framework for federated learning, offering scalability and efficiency without compromising accuracy in edge settings. The source code for our FedAGE is available at `https://anonymous.4open.science/r/FEDAGE-One-ShotFederatedLearning-832E`.

## 1 Introduction

Federated learning (FL) (McMahan et al., 2017; Kairouz et al., 2021) is a distributed machine learning paradigm in which numerous nodes, acting as clients, collaboratively train a shared global model under the coordination of a central server. Each client retains its raw data locally, transmitting only model updates to the server, which aggregates them to improve the global model. This setup always aims to ensure data privacy and maintain utility for clients. However, traditional FL methods (McMahan et al., 2017) require iterative communication between clients and the server. Consequently, they incur substantial communication overhead and elevate privacy risks, as repeated exchanges expand the temporal attack surface.

One-Shot Federated Learning (OFL) has emerged as a solution to these communication challenges by limiting model exchange to a single round. Naturally, this simplification comes at the cost of reduced model accuracy compared to iterative FL. Typical OFL approaches Guha et al. (2019) involve clients who train local models to completion and then transmit them to the server. The server aggregates these models by averaging (Guha et al., 2019), ensemble methods (Lin et al., 2020), or knowledge distillation (Zhu et al., 2021) using synthetic or public data into a final global model. Although these approaches mitigate communication costs, they often suffer from significantly reduced accuracy and high client-side computational demands. With the growing adoption of FL in edge environments, client-side computation has become a critical bottleneck. Resource-constrained edge devices are increasingly struggling with the training and communication demands of existing FL and OFL methods (Lim et al., 2020).

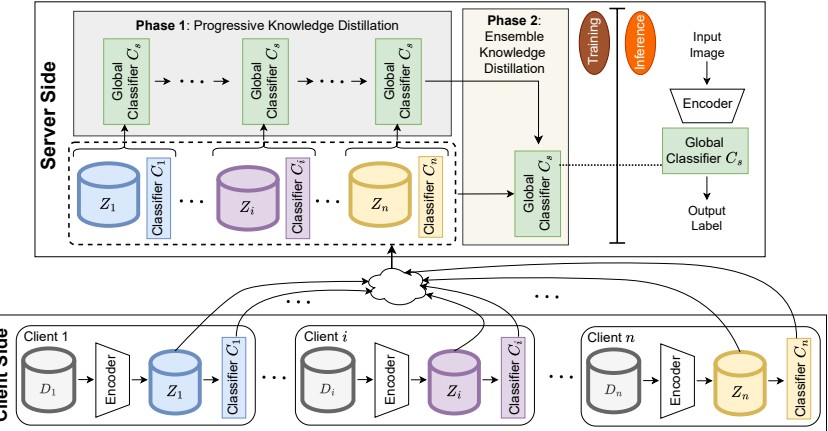

Figure 1: Illustration of the proposed FedAGE framework, where clients $i = 1, 2, \ldots, n$ share latent representations $Z_i$ and partial models $C_i$ (a set of fully-connected layers presenting the local classifiers) *only once*. A pretrained encoder is frozen and shared among all the clients and the server. The server maintains a global classifier $C_s$ to distill knowledge from clients during its training in two phases. During inference, the input's feature embedding, which is determined from the shared encoder, is sent to $C_s$ for classification.

In this work, we propose **FedAGE (Federated Aggregation of Generalized Embeddings)**, a novel federated learning framework (see Fig. 1) in which clients share latent representations instead of raw data or full model weights *only once*. These representations are generated using a shared frozen encoder, which can be pre-trained on public or auxiliary data. In addition, the weights of the classification (fully connected) layers from each trained client model are also shared with the server. This design drastically reduces client-side computation while preserving the discriminative structure required for effective global model building. By offloading the majority of the computations to the server and operating in a latent space, FedAGE improves both scalability and efficiency. The proposed FedAGE opens a promising direction for scalable and efficient FL, particularly suited for edge-centric applications where computational resources are constrained.

**Contributions.** 1) We introduce FedAGE, a novel one-shot federated learning framework where clients exchange latent representations rather than raw data and partial models instead of entire models. 2) We employ a shared, frozen encoder architecture to drastically reduce client-side computation and memory requirements while preserving discriminative features in the shared representations. Our method offloads all heavy computation to the server. 3) Through extensive experiments, we demonstrate that FedAGE outperformed state-of-the-art methods by a significant margin with reduced computation overhead, making it ideal for edge-device deployment. Further, it is shown that our FedAGE's performance is consistent across any level of data heterogeneity among clients.

## 1.1 RELATED WORKS

One-Shot Federated Learning (OFL) has emerged as a promising solution to the communication bottlenecks inherent in traditional federated learning approaches. The seminal work by (Guha et al., 2019) introduced the concept of one-shot FL, which constrains client-server communication to a single round, significantly reducing communication overhead while attempting to preserve model performance. This approach fundamentally differs from iterative FL methods by trading multiple communication rounds for a one-time exchange of trained models.

Guha *et al.* introduced the concept of OFL through two main approaches: heuristic ensemble client selection and knowledge distillation-based aggregation using auxiliary datasets (Guan et al., 2020). However, these early methods faced significant limitations in handling data heterogeneity and often required large public datasets for effective knowledge distillation. To address these challenges, advanced approaches have evolved, including synthetic data generation using generative networks (Zhu et al., 2021) and ensemble aggregation techniques (Lin et al., 2020). The state-of-the-art Co-Boosting algorithm (Dai et al., 2024) iteratively refines synthetic data and ensemble models.

Integration of foundation models has also shown promise in OFL. For example, FedPFT (Peng et al., 2024) demonstrates that large pretrained models can achieve performance comparable to multi-round FL in a single communication round. Despite these advances, OFL methods still continue to exhibit an accuracy gap compared to iterative FL, particularly under high data heterogeneity conditions. This motivates the development of more sophisticated aggregation strategies and privacy-preserving techniques, which we addressed in this work.

Knowledge Distillation (KD) (Gou et al., 2021) is a key technique in model compression that involves transferring insights from a larger, more complex "teacher" model to a smaller, efficient "student" model. KD (Hinton et al., 2015) was introduced in 2015. It has now emerged as a key approach for implementing high-performance models on devices with limited resources, all while preserving competitive levels of accuracy. In the context of OFL, KD plays an essential role by allowing the central server to combine various client models in a single communication round, eliminating the need to share raw data. Instead of simply aggregating the complete weights of the models, knowledge is effectively conveyed through "dark knowledge", which consists of soft predictions or distilled representations((Li & Wang, 2019),(Lin et al., 2020)).

Several other approaches (Lin et al., 2020), (Li & Wang, 2019) relied on auxiliary public datasets to facilitate knowledge transfer from client models to the server model, but these often faced challenges due to data heterogeneity and the limited availability of appropriate auxiliary datasets. Recently, there has been a significant shift towards employing generative models that can create synthetic data on the server side, which serves as a more effective medium for knowledge distillation, yielding promising results (Zhu et al., 2021), (Zhang et al., 2022). On the contrary, the proposed OFL setup FedAGE offers several key advantages over these federated learning approaches: 1) *Computational Efficiency* – Clients avoid the costly retraining of deep encoders, lowering computation and energy demands while achieving scalability across heterogeneous edge environments. 2) *Mitigation of Catastrophic Forgetting* – Through progressive distillation, ensemble strategies, and knowledge mixing, FedAGE effectively preserves and integrates knowledge across rounds. 3) *Improved Generalization* – The use of generalized embeddings allows for knowledge federation across clients' data distributions, improving performance under heterogeneous non-IID settings. 4) *Scalability in Edge Environments* – The lightweight client workload makes FedAGE more practical for deployment on edge devices with limited compute and memory resources.

## 2 PROBLEM DEFINITION

Federated learning is a decentralized machine learning paradigm that enables multiple clients to collaboratively train a unified global model while preserving the privacy of their raw data. Clients independently train local models on their own datasets and share only updates with the server, which aggregates them into a global model.. Until now, typical federated learning approaches merge local models via FedAvg (McMahan et al., 2017) to produce the global model.

Let us consider a federated learning system consisting of $n$ clients, denoted as $C_1, C_2, \ldots, C_n$, each possessing local datasets $D_1, D_2, \ldots, D_n$. In a dataset $D_i$, a sample is represented as a pair $(x, y)$ of an image $x$ and a corresponding class label $y$. The global objective function for the federated learning problem is mathematically formulated as $F(\theta) = \frac{1}{n} \sum_{i=1}^{n} F_i(\theta)$, where $F_i(\theta)$ represents the local objective function in the client $i$ as $F_i(\theta) = \frac{1}{|D_i|} \sum_{(x,y) \in D_i} \ell(\theta; x, y)$, where $\theta$ denotes the model parameters, $\ell(\cdot)$ is the loss function, and $|D_i|$ is the size of the local dataset at client $i$. Unlike traditional federated learning with multiple communication rounds between clients and server, the proposed OFL method operates under a one-shot learning constraint. where $T_{\text{comm}}$ represents the number of communication rounds between clients and the server.

### 2.1 NON-IID DATA DISTRIBUTION

In realistic federated learning scenarios, client data exhibits non-independent and identically distributed (non-IID) characteristics. We model this heterogeneity using a Dirichlet distribution (Hsu et al., 2019) to partition the global dataset across clients. Specifically, for each class $k$ at client $i$, we sample proportions $p_{i,k} \sim \text{Dir}(\alpha)$, where $\alpha$ is the concentration parameter controlling the degree of heterogeneity. When $\alpha \to \infty$, the distribution approaches IID, while $\alpha \to 0$ results in highly skewed, non-IID distributions. This formulation allows us to systematically control the statistical

heterogeneity across clients, where smaller $\alpha$ values indicate stronger non-IID characteristics. The data distribution at client $i$ for class $k$ is characterized by: $n_{i,k} = \lfloor p_{i,k} \cdot N_k \rfloor$, where $N_k$ is the total number of samples for class $k$ in the whole dataset $D_i$. More details and visualizations of non-iid data splits for different values of $\alpha$ are given in Appendix A.1.

# 3 FEDAGE: FEDERATED AGGREGATION OF GENERALIZED EMBEDDINGS

This section describes our proposed FedAGE, an OFL method based on Federated Shared Aggregation of Generalized Embeddings. The overall block diagram of our FedAGE is demonstrated in Fig. 1. FedAGE consists of three primary steps (see Fig. 1), such as latent embedding estimation and latent feature-driven classification at the client-side, and progressive knowledge distillation at the server side, as discussed in the following.

## 3.1 LATENT EMBEDDING ESTIMATION

On the client side, the latent embedding of an image $x_{r \times c}$ is obtained using a convolutional backbone. This backbone, consisting of convolutional blocks from a pre-trained network, is shared across all clients and the server. This backbone serves as an encoder that generates the latent embedding of $x$. In FedAGE, we keep this encoder frozen to reduce the clients' overhead from retraining, thereby improving computational efficiency and minimizing GFLOPS. We employ a frozen pre-trained encoder $E(\cdot; \theta_E) : \mathbb{R}^{r \times c} \to \mathbb{R}^d$ that maps input data from the original feature space to a lower-dimensional latent representation. The encoder parameters $\theta_E$ remain fixed throughout the proposed federated learning process. This design choice is motivated by the hypothesis that pretrained encoders capture universal latent feature representations that generalize across different data distributions.

## 3.2 CLIENT-SIDE LATENT REPRESENTATION LEARNING

Each client $i$ transforms all the samples of its local dataset $D_i$ into latent representations using the shared frozen encoder as $z_j = E(x_j)$ for each $x_j \in D_i$, which constitutes a set of latent representations $Z_i = \{z_j\}$. The latent dataset at client $i$ can then be defined as:

$$\mathcal{D}_i = \{(z_j, y_j) : z_j = E(x_j), \forall (x_j, y_j) \in D_i\}$$

Using these latent representations, each client trains a local (fully connected) classifier $C_i(\cdot) : \mathbb{R}^d \to \mathbb{R}^k$, where $k$ is the number of classes. Therefore, the client-specific optimization problem becomes:

$$\theta_{C_i}^* = \arg\min_{\theta_{C_i}} \frac{1}{|\mathcal{D}_i|} \sum_{(z,y) \in \mathcal{D}_i} \ell(C_i(z; \theta_{C_i}), y),$$

where $\theta_{C_i}$ represents the trainable parameters of $C_i(\cdot)$ and $\ell$ is *cross-entropy loss* in our implementation. Algorithm 2 in Appendix A outlines this procedure. The design choice of training only the client-specific classifier $C_i$ while keeping the shared encoder $E(\cdot)$ frozen is central to the method of FedAGE. The shared frozen encoder guarantees that all clients produce embeddings in a common latent space, ensuring comparability across different clients and heterogeneous data distributions, while at the same time reducing the client-side computation because only a small classifier is being trained and shared with the server. The client classifier $C_i$ captures decision boundaries in this shared latent space. The server receives latent embeddings and the client classifier, which acts as a teacher to distill knowledge into the server classifier.

## 3.3 SERVER-SIDE FEDERATED KNOWLEDGE AGGREGATION

Once the server receives the set of latent embeddings $Z_i$ generated from the local dataset and the trained classifier from clients, it focuses on building the *global model* (say, $M_s(\cdot)$) via federated knowledge aggregation at the server end. The server is built with the frozen shared encoder, $E(\cdot)$ and a fully connected global classifier $C_s(\cdot)$. FedAGE is classifier-architecture-agnostic: clients and the server may use different classifier heads in principle, see Appendix B.2 for further details. Thus, once $C_s(\cdot)$ is trained via federated aggregation, the prediction for any input image $x$ is obtained by combining the frozen encoder with the trained server classifier as:

$$M_s(x) = C_s(E(x))$$

---

**Algorithm 1** FedAGE: Server-Side Knowledge Distillation

---

1: **for** each client $i$ (sequentially) **do**  ▷ Teacher = $C_i$, Student = $C_s$, clients data embeddings = $Z_i$
2:     **for** each embedding $z \in Z_i$ **do**
3:         Compute pseudo labels: $\hat{y} = \arg\max P_i(z), \quad P_i(z) = \texttt{softmax}(C_i(z)/\tau)$
4:     **end for**
5:     Compute hard loss: $\mathcal{L}_{\text{hard}}^{(i)} = \frac{1}{|Z_i|} \sum_{z \in Z_i} \mathcal{L}_{\text{CE}}(\hat{y}, C_s^{(i)}(z))$
6:     Set weight for loss, $w_z = \max(P_i(z))$
7:     **if** $i > 1$ **then**
8:         Perform knowledge mixing as: $P_{\text{mixed}}(z) = \gamma P_{\text{teacher}}^{(i)}(z) + (1-\gamma) P_s^{(i-1)}(z)$
9:         Compute soft loss: $\mathcal{L}_{\text{soft}}^{(i)} = \frac{1}{|Z_i|} \sum_{z \in Z_i} w_z \cdot \texttt{KL}(P_{\text{mixed}}(z), P_s^{(i)}(z))$
10:     **else**
11:         Compute soft loss: $\mathcal{L}_{\text{soft}}^{(i)} = \frac{1}{|Z_i|} \sum_{z \in Z_i} w_z \cdot \texttt{KL}(P_i(z), P_s^{(i)}(z))$
12:     **end if**
13: **end for**
14: Optimize the knowledge distillation objective function: $\mathcal{L}_{\text{KD}}^{(i)} = \beta \cdot \mathcal{L}_{\text{hard}}^{(i)} + (1-\beta) \cdot \mathcal{L}_{\text{soft}}^{(i)}$
15: Perform ensemble distillation over all clients as: $P_{\text{ensemble}}(z) = \frac{1}{n} \sum_{i=1}^{n} P_i(z)$
16: Optimize the *ensemble loss* function: $\mathcal{L}_{\text{ensemble}} = \frac{1}{\sum_i |Z_i|} \sum_{z \in \cup Z_i} \texttt{KL}(P_{\text{ensemble}}(z), P_s^{(i+1)}(z))$
17: $C_s(\cdot)$ is the trained server-side classifier model in the global model $M_s(\cdot)$

---

This architecture enables the server to make predictions on new data while leveraging the universal feature representations learned by the pretrained encoder and the aggregated knowledge from all client classifiers. Altogether, these make the proposed OFL setup, FedAGE, unique by offering computational efficiency, improved generalization, and scalability in edge environments. However, in FedAGE, federated knowledge aggregation follows a two-step procedure, as described in the following subsections. Note that, in the rest of this section, client $i$ denotes the collection of the latent dataset $Z_i$ and the local classifier $C_i$, which are transmitted to the global server from client $i$.

### 3.3.1 PROGRESSIVE KNOWLEDGE DISTILLATION FRAMEWORK

We propose a *progressive knowledge distillation* approach to sequentially learn from past rounds of the global classifier, $C_s(\cdot) : \mathbb{R}^d \to \mathbb{R}^k$. Knowledge distillation (KD) (Hinton et al., 2015) is a well-established technique for transferring knowledge via model outputs. The actual ground truth labels for the latent embeddings, which are not transferred for privacy purposes, are thereby not available on the server end. Thus, we will formulate a student-teacher framework, where the client-side and server-side classifiers are considered as teacher and student models, respectively. In our progressive KD, first, the server (student) sequentially distils from each client's classifier (teacher) so that the server can accumulate client-specific decision behaviour. Then the student performs ensemble distillation to align it with the averaged information across teachers. Sequential learning allows effective knowledge transfer without label sharing (phase 1), while the ensemble step (phase 2) reduces bias caused by non-IID client data, as explained next.

**Phase 1: Client-Specific Sequential Distillation.** In this phase, the global classifier is trained over multiple rounds sequentially across all clients. For each client $i$, the server performs knowledge distillation using confidence-weighted soft distillation. To improve FedAGE's robustness, we minimize the difference in predictions by the student (server) and any teacher (client) with a weight. The weight is set to the teacher's confidence in the prediction, because high-confidence predictions influence the student more strongly while uncertain predictions contribute less. This confidence weighting is applied in both sequential and ensemble distillation. Thus, the loss is:

$$\mathcal{L}_{\text{KD}}^{(i)} = \beta \cdot \mathcal{L}_{\text{hard}}^{(i)} + (1-\beta) \cdot \mathcal{L}_{\text{soft}}^{(i)}, \tag{1}$$

where $\beta$ balances hard, $\mathcal{L}_{\text{hard}}^{(i)}$ and soft, $\mathcal{L}_{\text{soft}}^{(i)}$ losses, which are explained next. Larger $\beta$ places more weight on only pseudo-labels (referred to as *hard* labels) predicted by the teacher, while smaller $\beta$ emphasizes both pseudo-labels and the confidence scores (considered as *soft* labels).

*Hard Label Loss (Privacy-Preserving Pseudo Labels)*: Each input embedding $z \in Z_i$ for the client $i$ is sent to the teacher model to generate its pseudo class (hard) labels $\hat{y}$. Subsequently, the loss

between these hard labels and the student's output, which is termed as *hard loss*, is calculated as:

$$\mathcal{L}_{\text{hard}}^{(i)} = \frac{1}{|Z_i|} \sum_{z \in Z_i} \mathcal{L}_{\text{CE}}(\hat{y}, C_s^{(i)}(z)), \tag{2}$$

where $C_s^{(i)}$ denotes the server-side classifier, which is presently getting learned with the latent embedding of the client $i$ in the sequential training procedure, and $\mathcal{L}_{\text{CE}}$ is the *cross-entropy loss*.

*Soft Distillation Loss:* The hard distillation approach does not weigh the predictions obtained from the student, even though it is learning from the teacher for the first time. Note that the student model $C_s^{(i)}$ is being trained using data and the teacher model for client $i$. Let $P_i(z)$ and $P_s^{(i)}(z)$ denote the class probabilities from the teacher and student models, respectively. Each is calculated using the softmax function $\texttt{softmax}(C_i(z)/\tau)$, where $\tau$ is the *temperature* — a scaling factor that controls the smoothness of the softmax output (higher $\tau$ produces softer probability distributions, while lower $\tau$ makes distributions sharper). The class probabilities for both teacher and student are derived using the same $\tau = 2$, in our implementation. In FedAGE, the *soft distillation loss* includes a confidence-weighting term on the student model's predictions as:

$$\mathcal{L}_{\text{soft}}^{(i)} = \frac{1}{|Z_i|} \sum_{z \in Z_i} w_z \cdot \texttt{KL}(P_i(z), P_s^{(i)}(z)), \tag{3}$$

where $\texttt{KL}(\cdot, \cdot)$ is the *KL-divergence loss* and $w_z$ is the confidence weight. In our implementation, $w_z$ is set to the maximum of $P_i(z)$ to emphasize input embedding where the teacher has high confidence, improving knowledge transfer quality.

**Phase 2: Ensemble Distillation.** After sequential learning from individual clients i.e. teacher models, the server performs ensemble distillation using combined knowledge from all teachers $C_i, i = 1 \ldots n$ by optimizing the *ensemble loss* function as:

$$\mathcal{L}_{\text{ensemble}} = \frac{1}{\sum_i |Z_i|} \sum_{z \in \cup Z_i} \texttt{KL}(P_{\text{ensemble}}(z), P_s^{(i+1)}(z)), \tag{4}$$

where $P_{\text{ensemble}}(z) = \frac{1}{n} \sum_{i=1}^{n} P_i(z)$ denotes the average class probabilities across all teacher models, and $P_s^{(i+1)}(z)$ represents the probability predicted by the updated student model after completing $n$ cycles of sequential training over the $n$ client (teacher) models. This way, the server distils collective knowledge from all teachers, i.e., clients. Since client datasets are often heterogeneous, relying on a single teacher may bias the student model. To mitigate the bias in non-IID settings by optimizing against the ensemble distribution $P_{\text{ensemble}}(z)$, the student aligns with the average decision boundary across all teachers, reducing skew from data imbalance. Finally, the trained global server model $M_s$ is obtained with the trained server-side classifier $C_s$ for inference. The two-step knowledge distillation process is outlined in Algorithm 1.

### 3.3.2 KNOWLEDGE MIXING STRATEGY TO MITIGATE CATASTROPHIC FORGETTING

Sequentially distilling from clients can induce *catastrophic forgetting* (Li & Hoiem, 2017), (Kirkpatrick et al., 2017) of previously learned clients' knowledge, since the student is updated to match the performance of the current teacher at each step. To mitigate this, for clients $i > 1$, we employ a knowledge mixing strategy that combines the current teacher's knowledge with previously acquired knowledge as:

$$P_{\text{mixed}}(z) = \gamma \cdot P_{\text{teacher}}^{(i)}(z) + (1 - \gamma) \cdot P_s^{(i-1)}(z), \tag{5}$$

where $\gamma$ is a mixture coefficient, $P_{\text{teacher}}^{(i)}(z)$ represents the current teacher's (client $i$' s) softmax prediction, and $P_s^{(i-1)}(z)$ denotes the prediction from the server model $C_s^{(i-1)}(z)$ trained from the clients $1 \ldots (i-1)$. The mixture coefficient $\gamma$ controls the balance between new knowledge acquisition and knowledge retention. As $\gamma$ increases, the preservation of prior knowledge decreases. The *soft loss* (see Algorithm 1 and equation 3) integrates this $P_{\text{mixed}}(z)$ for knowledge mixing as:

$$\mathcal{L}_{\text{soft}}^{(i)} = \frac{1}{|Z_i|} \sum_{z \in Z_i} w_z \cdot \texttt{KL}(P_{\text{mixed}}(z), P_s^{(i)}(z)), \tag{6}$$

where $P_s^{(i)}(z)$ is the server's (student) updated prediction after learning from client $i$ (teacher).

## 4 EXPERIMENTS

In this section, we conduct a series of experiments to empirically validate the effectiveness of our OFL framework, FedAGE. Our evaluation is designed to address three primary questions: (a) How robust is FedAGE to statistically simulated data heterogeneity compared to OFL baselines? (b) How well does FedAGE scale with an increasing number of clients? (c) How much does FedAGE reduce the client-side computation overhead?

### 4.1 EXPERIMENTAL SETUP

**Datasets and partitions.** We evaluate our method on five widely used public image classification datasets: CIFAR-10 (Krizhevsky, 2009), CIFAR-100 (Krizhevsky, 2009), STL-10 (Coates et al., 2011), MNIST (LeCun et al., 2002), and FashionMNIST (Xiao et al., 2017). To simulate realistic FL scenarios, we distribute their training sets among clients in a non-independent and identically distributed (non-IID) manner, while we use their test set to determine the server's test accuracy. We use a Dirichlet distribution (see Appendix A.1) with a concentration parameter $\alpha$ to partition the data, as described in Section 2.1. We test across varying degrees of heterogeneity by setting $\alpha$ to $0.01, 0.05$, and $0.1$, where a smaller $\alpha$ indicates a higher degree of data skewness.

**Baselines.** We compare FedAGE against a set of recent OFL baselines: FedAVG (McMahan et al., 2017), FedKD (Gong et al., 2022), FedCVAE (Heinbaugh et al., 2023), and Co-Boosting (Dai et al., 2024). The results are reproduced with their publicly available source codes in our computing setup. Some recent FL methods such as FENS (Allouah et al., 2024), FEDGEN (Zhu et al., 2021), SCAFFOLD (Karimireddy et al.) and FEDPROX (Li et al., 2020) are inappropriate for comparison in our proposed OFL framework. The method FENS is a hybrid FL and OFL setup, whereas FEDGEN, SCAFFOLD and FEDPROX focus on other learning perspectives like regularization and are not compatible in a one-shot setting.

**Implementation Details.** For FedAGE, we adopt a ResNet-18 backbone (see Appendix B.1), pre-trained on ImageNet (Deng et al., 2009), as the shared frozen encoder $E(\cdot)$. Both client- and server-side classifiers, $C_i(\cdot)$ and $C_s(\cdot)$, are lightweight fully connected networks, detailed in Appendix B.2. Although Table 4 shows that FedAGE is classifier-architecture-agnostic, we employ a uniform classifier across all clients in our experiments. Results are reported as mean $\pm$ standard deviation of accuracy (%) over three random seeds. All models are trained using the Adam optimizer with $\beta = 0.5$ (see ablation study), while $\gamma$ varies by dataset (Appendix B.3). Dataset-specific hyperparameters are selected via grid search to maximize mean accuracy across seeds and heterogeneity levels ($\alpha$), as listed in Table 5 (Appendix B.4). Unless otherwise specified, the default setting is $n = 10$ clients with $\alpha = 0.01$ (extremely heterogeneous).

### 4.2 RESULTS AND ANALYSIS

**Robustness to Data Heterogeneity.** Our first set of experiments evaluates all methods under varying degrees of non-IID data distribution, with results summarized in Table 1. Across all datasets and heterogeneity levels, FedAGE consistently outperforms all baselines except in one case. On CIFAR-10, under the most challenging setting ($\alpha = 0.01$), FedAGE achieves 78.90% accuracy, compared to only 32.47% for the next best baseline, Co-Boosting. The performance gap narrows as data distributions become more balanced; for example, on CIFAR-100 with $\alpha = 0.1$, FedAGE attains 50.38% accuracy, exceeding FedKD by over 6%. These results demonstrate that sharing latent representations — embedding discriminative features — is a more effective strategy for knowledge aggregation than exchanging full model parameters or relying on generative distillation, especially under severe data heterogeneity. This trend is further illustrated in Fig. 2(a), where FedAGE exhibits minimal variance across heterogeneity levels, unlike baselines. Since parameter averaging is a commonly used aggregation strategy in federated learning, we also compare against both simple and weighted parameter averaging. While simple and weighted averaging attain below 26% test accuracy on CIFAR-10, Progressive KD consistently achieves close to 79%, representing a substantial improvement. Detailed results and discussion are provided in Appendix B.6.

**Scalability with Number of Clients.** We also consider the scalability of FedAGE as the number of participating clients increases. For this experiment, the heterogeneity level is fixed at $\alpha = 0.1$ on CIFAR-10, and the number of clients is varied as $n \in \{5, 10, 20, 50\}$. The results, shown in

Table 1: Test accuracy (%) of server models of different OFL methods for various heterogeneity levels $m$ across datasets. The rightmost column $\Delta$ shows the difference between FedAGE and the top-performing baseline. The best results for each $m$ and each dataset are in light blue.

| Dataset | $\alpha$ | FedAvg | FedKD | FedCVAE | Co-Boosting | **FedAGE** | $\Delta$ |
|---|---|---|---|---|---|---|---|
| | 0.01 | 11.84±2.00 | 18.86±2.01 | 29.55±0.45 | 32.47±0.44 | 78.90±0.82 | +46.4 |
| **CIFAR-10** | 0.05 | 12.84±1.86 | 37.96±4.82. | 28.67±1.15 | 39.23±1.06 | 78.72±0.57 | +39.4 |
| | 0.1 | 12.10±3.45 | 65.98±5.49 | 30.34±0.38 | 56.23±0.60 | 78.36±0.74 | +12.38 |
| | 0.01 | 1.08±0.14 | 29.23±4.36 | 06.90±0.17 | 14.71±2.17 | 48.88±0.61 | +19.6 |
| **CIFAR-100** | 0.05 | 1.06±0.12 | 41.12±0.77 | 07.14±0.53 | 20.09±0.86 | 49.52±0.42 | +8.4 |
| | 0.1 | 1.13±0.09 | 44.20±2.91 | 07.24±0.38 | 23.89±0.64 | 50.38±0.15 | +6.18 |
| | 0.01 | 14.64±6.30 | 82.16±1.95 | 86.15±0.47 | 90.67±0.43 | 93.12±0.66 | +2.4 |
| **MNIST** | 0.05 | 36.59±12.78 | 86.48±1.22 | 85.93±0.82 | 92.60±0.68 | 93.11±0.73 | +0.5 |
| | 0.1 | 45.77±10.67 | 92.89±0.93 | 86.47±0.23 | 93.90±1.01 | 92.96±0.42 | -0.9 |
| | 0.01 | 11.53±2.91 | 37.31±2.21 | 73.14±1.36 | 41.71±0.41 | 85.14±0.41 | +12 |
| **FashionMNIST** | 0.05 | 23.92±7.35 | 47.66±3.63 | 74.23±0.41 | 48.40±1.11 | 85.26±0.31 | +11.0 |
| | 0.1 | 39.75±8.58 | 71.34±1.55 | 72.49±0.66 | 67.64±1.70 | 85.05±0.39 | +12.5 |
| | 0.01 | 10.03±0.07 | 39.63±1.86 | 23.00±0.75 | 40.15±0.43 | 90.32±0.49 | +50.1 |
| **STL10** | 0.05 | 9.79±0.86 | 42.64±1.20 | 24.01±0.18 | 43.33±1.06 | 90.24±0.44 | +46.9 |
| | 0.1 | 11.46±1.40 | 60.42±2.73 | 25.63±0.21 | 61.98±1.22 | 90.12±0.72 | +28.1 |

Table 2: Test accuracy (%) of various OFL methods on CIFAR-10 over different number of clients ($n$). The best results for each $n$ are highlighted in light blue.

| $n$ | FedAvg | FedKD | FedCVAE | Co-Boosting | **FedAGE** | $\Delta$ |
|---|---|---|---|---|---|---|
| 5 | 18.32±7.18 | 71.47±3.01 | 27.10±1.01 | 55.33±0.89 | 78.23±0.57 | +6.7 |
| 10 | 13.29±2.07 | 65.98±5.49 | 30.34±0.38 | 56.23±0.60 | 78.36±0.74 | +12.3 |
| 20 | 12.07±2.54 | 64.30±4.17 | 32.63±0.55 | 50.37±1.44 | 78.28±0.37 | +13.9 |
| 50 | 10.02±0.04 | 60.14±2.218 | 33.18±0.12 | 44.73±2.08 | 78.28±0.68 | +18.1 |

Table 2, demonstrate that FedAGE consistently outperforms all baselines across client scales. While most methods suffer accuracy degradation as the number of clients grows — reflecting the increased challenge of aggregating over diverse models — FedAGE remains robust. With $n = 50$ clients, it achieves $78.28\%$ accuracy, exceeding the best baseline (FedKD) by more than $18\%$. These findings highlight FedAGE's scalability and its suitability for large-scale federated deployments on edge devices.

**Efficiency in Client-Side Computation.** We evaluate client-side computation on CIFAR-10 using two metrics: the number of trainable parameters and the giga floating-point operations (GFLOPs) required per client. Most OFL methods, such as Co-Boosting and FedCVAE, train a full model (ResNet-18 in this experiment) on each client before sharing, leading to substantial computational overhead. In contrast, FedAGE only requires a one-time forward pass through a frozen encoder (ResNet-18 convolutional block pretrained on ImageNet) to generate latent vectors, after which each client trains a lightweight classifier head. This design drastically reduces client-side computation, making FedAGE particularly well-suited for edge environments. Specifically, each client in FedAGE trains only a lightweight classifier head with *9,610 trainable parameters*, compared to *11.2 million parameters* in a full ResNet-18. This corresponds to a reduction of over *1000×* in client-side trainable parameters (see Fig. 2(b)), underscoring FedAGE's computational efficiency.

We evaluate client-side computation in terms of GFLOPs per training sample. Training a full ResNet-18 requires $\sim 5.4$ GFLOPs per sample, whereas FedAGE needs only a frozen encoder forward pass ($\sim 1.8$ GFLOPs/sample) and lightweight classifier training ($\sim 5.7 \times 10^{-5}$ GFLOPs/sample), yielding a $\sim 66.6\%$ reduction. Unlike baselines that repeatedly incur the full ResNet-18 cost in every epoch, FedAGE requires the encoder only once, after which the per-sample cost converges to nearly zero (see Fig. 2(c)). This makes FedAGE highly efficient for resource-constrained edge devices, see Appendix B.7 for more details.

**Ablation Study on the Loss Components** We conduct an ablation study to assess the individual contributions of the *hard* and *soft* distillation losses, controlled by the hyperparameter $\beta$ in FedAGE's

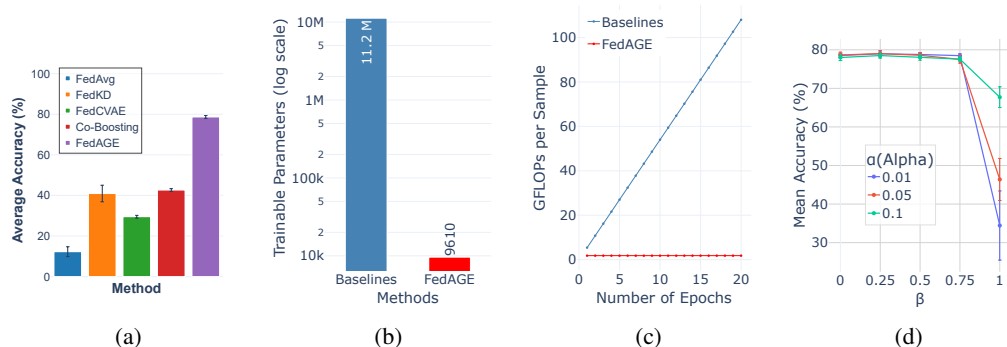

Figure 2: (a) Error bar plot showing accuracy vs. data heterogeneity ($\alpha$), (b) Bar plot showing number trainable parameters for FedAGE and baselines, (c) Line plot for GFLOPs required per sample vs. epoch plot, and (d) Plot for accuracy vs. loss balancing parameter $\beta$.

total loss function (equation 1). As shown in Figure 2(d), model accuracy remains consistently high for $\beta \in [0, 0.75]$ across all data heterogeneity levels ($\alpha$), highlighting the effectiveness of the soft-label distillation loss $\mathcal{L}_{soft}$ for knowledge transfer. In contrast, setting $\beta = 1$ — relying solely on hard pseudo-labels — causes a sharp performance drop, especially under severe non-IID conditions ($\alpha = 0.01$). This underscores the necessity of soft, confidence-weighted teacher signals, as hard labels alone are insufficient for robust distillation in federated settings. Nevertheless, $\mathcal{L}_{hard}$ provides an unambiguous supervisory signal that regularizes training by enforcing commitment to the teacher's most confident predictions. The best performance, observed for $0.25 < \beta < 0.75$, demonstrates that combining nuanced (soft) and decisive (hard) supervision yields the most reliable outcomes.

**Privacy Analysis**   The architectural design of FedAGE provides strong inherent privacy advantages. By sharing only latent embeddings from a frozen, pre-trained encoder, the method avoids vulnerabilities of gradient sharing, such as gradient inversion attacks. This privacy-by-design approach is theoretically motivated by the *Information Bottleneck Principle* (Tishby et al., 2000), (Tishby & Zaslavsky, 2015), which ensures that shared embeddings retain task-relevant information while discarding unnecessary details about the raw inputs. For scenarios demanding formal guarantees, FedAGE can be extended with *local Differential Privacy (DP)*. An analysis in Appendix C outlines the theoretical underpinnings, demonstrates empirical resilience against reconstruction attacks, and examines the privacy–utility trade-off under DP.

**Limitations and Future Work.** While FedAGE achieves notable gains in computational efficiency and accuracy for one-shot federated learning, a couple of limitations remain that open avenues for future research. Its performance depends heavily on the quality and domain alignment of the frozen encoder; significant mismatches with client data can yield suboptimal embeddings and limit global accuracy. Second, although FedAGE reduces client-side costs, it shifts computational and memory demands to the server, which must store all latent embeddings and execute the distillation process, potentially creating bottlenecks at scale. We plan to address these issues and propose to extend FedAGE to tackle more complex tasks, such as federated object detection, in future work.

## 5   CONCLUSION

In this work, we introduced FedAGE, a one-shot federated learning framework that reduces client-side computation by sharing latent representations and a lightweight classifier head instead of a full model. Through progressive distillation with confidence-weighted soft labels, ensemble distillation, and knowledge mixing, FedAGE mitigates catastrophic forgetting and improves generalization under data heterogeneity. Experiments on five benchmarks show that FedAGE consistently surpasses state-of-the-art baselines in accuracy and efficiency, while remaining robust under severe non-IID settings and scalable with more clients. These results highlight FedAGE as a practical solution for resource-constrained edge environments.

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

# A ADDITIONAL DETAILS ON METHODS

## A.1 MORE DETAILS ON DIRICHLET PARTITIONING

The **Dirichlet distribution**, denoted as $\text{Dir}(\alpha)$, is a probability distribution over a set of positive numbers that sum to one. It is conrolled by a single parameter, $\alpha > 0$. In the context of Federated Learning, we leverage this to control the statistical heterogeneity of the data distribution across clients. The partitioning process works as follows: for each class $k$, we sample a probability vector $\mathbf{p}_k = (p_{1,k}, p_{2,k}, \ldots, p_{n,k}) \sim \text{Dir}(\alpha)$, where $n$ is the number of clients. This vector determines the proportion of samples of class $k$ that is allocated to each client $i$.

The concentration parameter $\alpha$ is crucial for controlling the degree of non-IID-ness. A small $\alpha$ (e.g., $\alpha \to 0$) results in partitions where the probability mass in the vector $\mathbf{p}_k$ is concentrated on only a few clients. This leads to a highly skewed, non-IID distribution, where each client possesses a large number of samples from only one or a very small number of classes. A large $\alpha$ (e.g., $\alpha \to \infty$) results in partitions where the proportions in $\mathbf{p}_k$ are similar for all clients ($p_{k,1} \simeq p_{k,2} \simeq \cdots \simeq p_{k,C}$). This mimics an IID setting, where each client receives a balanced distribution of data from all classes.

### A.1.1 VISUALIZATION OF DATA SKEWNESS

Figure 3 provides a clear visual representation of this effect on the CIFAR-10 dataset for its 10 classes distributed across 10 clients. In each heatmap for each of $\alpha \in \{0.01, 0.05, 0.1\}$, the rows correspond to client IDs and the columns to class IDs. The color and numerical value in each cell $(i, k)$ represent the number of samples of class $k$ held by client $i$.

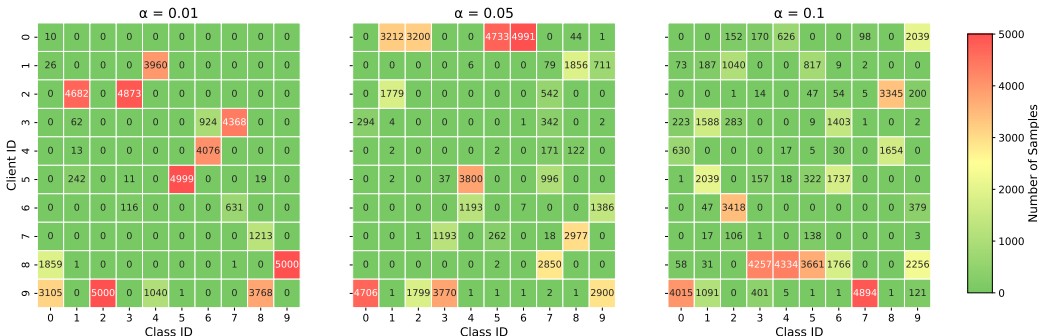

Figure 3: Comparison of data distribution across 10 clients for CIFAR-10 across its 10 classes under a Dirichlet partition with varying $\alpha$ values 0.1, 0.5, and 1.0).

With $\alpha = 0.01$, we observe extreme data heterogeneity. The distribution is sparse, with most clients holding a significant majority of samples from just a single class. For example, Client 8 possesses nearly 5,000 samples of Class 4 and almost none from other classes. Increasing to $\alpha = 0.05$, the distribution becomes slightly more balanced, though still highly skewed. Clients now begin to hold non-trivial amounts of data from multiple classes, but a strong class imbalance per client persists. At $\alpha = 0.1$, the trend towards a more uniform distribution continues. The data is more spread out across the client-class matrix, representing a moderate level of non-IID heterogeneity.

This systematic approach allows us to precisely control and simulate the varying degrees of data skew found in real-world decentralized datasets, providing a framework for evaluating the performance of our one-shot federated learning method under different heterogeneity conditions.

### A.1.2 CLIENT-SIDE TRAINING ALGORITHM

The entire client-side training procedure is mathematically explained in Section 3.2 of the main paper. In summary, each client in FedAGE first converts its local dataset into latent representations using a shared frozen encoder. This process produces a latent dataset where each input sample is paired with its corresponding label. Instead of training a full model, the client then trains only a lightweight classifier on these latent representations. Here, Algorithm 2 summarizes the entire client-side training procedure step-wise.

---

**Algorithm 2** FedAGE: Client-Side Training

---

1: Initialize frozen encoder $E(\cdot; \theta_E)$ with fixed parameters $\theta_E$
2: **for** each client $i \in \{1, \ldots, n\}$ **do**
3:     Transform local dataset into latent space: $\mathcal{D}_i = \{(z_j, y_j) : z_j = E(x_j), \forall (x_j, y_j) \in D_i\}$
4:     Train local classifier $C_i$: $\theta_{C_i}^* = \arg\min_{\theta_{C_i}} \frac{1}{|\mathcal{D}_i|} \sum_{(z,y) \in \mathcal{D}_i} \ell(C_i(z; \theta_{C_i}), y)$
5:     Transfer $C_i$ and $Z_i = \{z_j \in \mathcal{D}_i\}$ to server
6: **end for**

---

# B    Additional Details on Experimental Results and Analysis

## B.1    Results with Different Backbone Architectures

To evaluate the generalizability and robustness of our proposed one-shot federated learning algorithm, we conducted experiments using the convolution block of several distinct backbone neural network architectures as encoders. This analysis is crucial to ensure that the effectiveness of our method is not limited to a single model type but can be applied more broadly. We tested four popular architectures: ResNet18 (He et al., 2016), the lightweight MobileNet (Howard et al., 2017) and SqueezeNet (Iandola et al., 2016), and VGG19 (Simonyan & Zisserman, 2014). The experimental setup remained consistent with our main results, i.e. we have used publicly available versions of the architectures, pretrained on ImageNet.

The results of this comparative analysis are presented in Table 3. The primary observation is the strong and consistent performance of our method when paired with modern architectures. ResNet18 consistently achieves the highest test accuracy, reaching up to $78.79\% \pm 0.52\%$ even under the most severe data skew ($\alpha = 0.01$). The MobileNet architecture also performs well, with its accuracy closely trailing that of ResNet18. A critical finding is the algorithm's robustness across all tested levels of non-IID data. For these models, there is minimal degradation in accuracy as $\alpha$ decreases from 0.1 to 0.01. For instance, ResNet18's mean accuracy only varies by about 0.7% across the entire range, highlighting that our one-shot aggregation mechanism is highly effective at handling statistical heterogeneity. While ResNet18 and MobileNet are the top performers, the algorithm still achieves reasonable accuracy with SqueezeNet. VGG19 shows comparatively lower performance, suggesting that certain architectures may be less suited to the one-shot aggregation process on this task, although the method still provides stable convergence.

In conclusion, these results strongly suggest that our proposed algorithm is largely **model-agnostic** and is particularly effective when integrated with modern, efficient convolutional neural networks. The stable performance across different architectures and severe data distributions underscores the general applicability and robustness of our approach.

## B.2    Classifier-Agnostic Results

Beyond statistical data heterogeneity, a key challenge in federated learning is **system heterogeneity**, where clients may employ different model architectures due to varying hardware capabilities or software versions. To demonstrate the robustness of our proposed method to this challenge, we conducted an extensive analysis using classifiers with varying architectural depths. This experiment is designed to show that our algorithm can effectively aggregate knowledge even when client models are not identical.

We defined three distinct classifier architectures for clients, all of which are simple multi-layer perceptrons (MLPs) that attach to the shared frozen encoder: **Classifier 1** is a 2-layer MLP (Input:

Table 3: Test accuracy (%) of different backbone models under various Dirichlet partitioning levels ($\alpha$) on CIFAR-10 for 10 classes with 10 clients.

| $\alpha$ | ResNet18 | MobileNet | SqueezeNet | VGG19 |
|---|---|---|---|---|
| 0.01 | **78.79** $\pm$ 0.52 | 76.87 $\pm$ 0.23 | 74.86 $\pm$ 0.59 | 58.20 $\pm$ 0.00 |
| 0.05 | **78.58** $\pm$ 0.72 | 76.89 $\pm$ 0.41 | 74.60 $\pm$ 0.72 | 55.16 $\pm$ 0.00 |
| 0.10 | **78.05** $\pm$ 0.71 | 76.30 $\pm$ 0.08 | 73.73 $\pm$ 0.41 | 55.69 $\pm$ 1.06 |

$64 \rightarrow 128 \rightarrow$ Output: number of classes), **Classifier 2** is a 3-layer MLP (Input: $64 \rightarrow 128 \rightarrow 32 \rightarrow$ Output: number of classes), and **Classifier 3** is a 4-layer MLP (Input: $64 \rightarrow 128 \rightarrow 32 \rightarrow 16 \rightarrow$ Output: number of classifier). We then created three experimental scenarios: (i) a **homogeneous** setting where all clients use Classifier 1 (*notably*, we use this setup in all of our experiments, unless otherwise specified); (ii) a **heterogeneous** setting where clients alternate between Classifier 1 and 2; and (iii) a more **diverse** setting where clients cycle through all three classifier architectures. These experiments were run across a range of client numbers, 5, 10, 20, and 50, and the data skew levels with $\alpha \in \{0.01, 0.05, 0.1\}$, while keeping the frozen encoder backbone and other hyperparameters constant.

The results, presented in Table 4, show that our algorithm's performance is remarkably stable and largely independent of the architectural diversity among client classifiers. For any given number of clients and $\alpha$ value, the test accuracy remains consistently high whether one, two, or three different classifier architectures are used. For instance, with 50 clients and severe data skew ($\alpha = 0.01$), the accuracies are $78.87\%$, $78.85\%$, and $78.68\%$ for one, two, and three classifiers, respectively—a negligible difference. This consistency demonstrates that our method can effectively fuse knowledge from architecturally distinct models without a performance penalty.

This analysis confirms that our method is **classifier-agnostic**. This is a significant advantage for real-world federated systems, as it removes the strict constraint that all participating clients must maintain identical model architectures, enhancing the flexibility and practicality of deployment.

Table 4: Test accuracy (5) of the server model while varying the client-side classifiers across different numbers of clients and different $\alpha$ values.

| $\alpha$ | No. of Classifiers | Number of Clients ($n$) | | | |
|---|---|---|---|---|---|
| | | 5 | 10 | 20 | 50 |
| | 1 | $78.07 \pm 0.26$ | $78.79 \pm 0.52$ | $78.80 \pm 0.47$ | $78.87 \pm 0.47$ |
| 0.01 | 2 | $78.10 \pm 0.41$ | $78.80 \pm 0.50$ | $78.64 \pm 0.59$ | $78.85 \pm 0.57$ |
| | 3 | $78.03 \pm 0.37$ | $78.91 \pm 0.49$ | $78.51 \pm 0.54$ | $78.68 \pm 0.53$ |
| | 1 | $77.96 \pm 0.51$ | $78.58 \pm 0.72$ | $78.82 \pm 0.69$ | $78.47 \pm 0.42$ |
| 0.05 | 2 | $78.28 \pm 0.75$ | $78.51 \pm 0.67$ | $78.75 \pm 0.46$ | $78.40 \pm 0.56$ |
| | 3 | $77.96 \pm 0.78$ | $78.66 \pm 0.54$ | $78.60 \pm 0.66$ | $78.36 \pm 0.63$ |
| | 1 | $77.98 \pm 0.75$ | $78.05 \pm 0.71$ | $78.37 \pm 0.58$ | $78.32 \pm 0.33$ |
| 0.1 | 2 | $78.04 \pm 0.66$ | $78.11 \pm 0.80$ | $78.40 \pm 0.71$ | $78.37 \pm 0.63$ |
| | 3 | $78.33 \pm 0.72$ | $77.93 \pm 0.97$ | $78.30 \pm 0.55$ | $77.95 \pm 0.59$ |

## B.3 VARIATION OF ACCURACY WITH GAMMA

The *mixture coefficient*, $\gamma$, introduced in equation 5, is a key hyperparameter that balances new knowledge acquisition against the retention of prior knowledge to mitigate catastrophic forgetting. We performed a detailed study to analyze the model's sensitivity to $\gamma$ by evaluating its performance for $\gamma \in \{0, 0.25, 0.5, 0.75, 1.0\}$ across various data heterogeneity settings, $\alpha \in \{0.01, 0.05, 0.1\}$ for the CIFAR-10 dataset.

The results, shown in Figure 4, demonstrate the efficacy of our knowledge mixing strategy. The configuration with $\gamma = 0$, which entirely discards new client information, consistently yields the worst performance. Conversely, we observe peak performance at $\gamma = 0.25$ for the more challenging non-IID scenarios ($\alpha = 0.01$ and $\alpha = 0.05$), indicating that placing a greater emphasis on knowledge retention while incorporating new knowledge is optimal. Although the model exhibits robustness to the choice of $\gamma$ for all values greater than zero, this study validates that a carefully balanced mixture is superior to standard sequential distillation ($\gamma = 1.0$).

## B.4 HYPERPARAMETER SETTINGS

The hyperparameters setting for different datasets can be found in Table 5. These have been found by performing a grid search on these hyperparameters. Other than these, batch size is fixed at 64, optimizer is Adam, and $\beta = 0.5$ for all datasets.

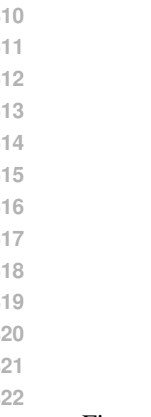

Figure 4: Mean accuracy as a function of the knowledge mixture coefficient $\gamma$. Each line represents a different degree of non-IID data distribution ($\alpha \in \{0.01, 0.05, 0.1\}$). The results show that a balanced mixture of old retention and new knowledge achieves the highest performance.

Table 5: Best hyperparameters for different datasets.

| Dataset | Learning Rate | $\gamma$ | Epochs |
|---|---|---|---|
| CIFAR-10 | 0.001 | 1.0 | 7 |
| CIFAR-100 | 0.005 | 0.5 | 7 |
| STL-10 | 0.005 | 1.0 | 7 |
| MNIST | 0.001 | 0.75 | 7 |
| Fashion-MNIST | 0.001 | 0.75 | 7 |

### B.5 SYSTEM CONFIGURATIONS

The primary computational system used for the experiments was equipped with two Intel® Xeon® Gold 6426Y processors, providing a total of 32 cores running at 2.50 GHz with a 57.5M cache. For graphical and parallel processing, the system utilized two NVIDIA RTX 6000 Ada Generation GPUs, each with 48 GB of GDDR6 ECC memory. The system was configured with 512 GB of DDR5 3200 MHz Registered ECC RAM.

### B.6 COMPARISON WITH PARAMETER AVERAGING

We also compare our server-side *Progressive Knowledge Distillation* (Progressive KD) approach with standard parameter averaging techniques. Specifically, we evaluate two common variants: *Simple Averaging*, which computes the arithmetic mean of parameters across all client models, and *Weighted Averaging* (McMahan et al., 2017), which assigns weights to client models proportional to their dataset sizes. As shown in Table 6, Progressive KD consistently outperforms both parameter averaging strategies by a large margin across all settings of $\alpha$. Moreover, parameter averaging methods are inherently limited to scenarios where all client models share the same architecture, whereas our approach is agnostic to the choice of client classifiers (see Appendix B.2).

Table 6: Test accuracy (%) of the proposed FedAGE integrating different KD approaches (Simple Averaging, Weighted Averaging, and our Progressive KD) on CIFAR-10. The best results for each $n$ are highlighted in light blue .

| $\alpha$ | Simple Averaging | Weighted Averaging | **Progressive KD (Ours)** |
|---|---|---|---|
| 0.01 | $15.08 \pm 3.72$ | $19.50 \pm 4.32$ | $78.90 \pm 0.82$ |
| 0.05 | $13.66 \pm 5.17$ | $14.93 \pm 6.63$ | $78.72 \pm 0.57$ |
| 0.1 | $12.66 \pm 3.41$ | $25.75 \pm 10.71$ | $78.36 \pm 0.74$ |

### B.7 Calculation of GFLOPs

To compute GFLOPs per training sample, all experiments resize CIFAR-10 images to $224 \times 224$ to match the ResNet-18 configuration. A forward pass through ResNet-18 costs $\sim 1.8$ GFLOPs per image (Albanie, 2025), and using the conservative estimate training GFLOPs $\approx$ GFLOPs$_{\text{forward}}$ + GFLOPs$_{\text{backward}}$, with backward propagation costing roughly twice the forward pass (Hobbhahn & Sevilla, 2021; OpenAI, 2018), training a full ResNet-18 requires $\sim 5.4$ GFLOPs per sample. In FedAGE, clients only perform one forward pass through the frozen ResNet-18 encoder ($\sim 1.8$ GFLOPs/sample) and the classifier head ($\sim 1.9 \times 10^{-4}$ GFLOPs/sample), with backpropagation limited to the classifier head ($\sim 3.8 \times 10^{-4}$ GFLOPs/sample), yielding a negligible classifier cost of $\sim 5.77 \times 10^{-5}$ GFLOPs per sample. Thus, FedAGE requires only $\sim 1.8$ GFLOPs per sample compared to $\sim 5.4$ GFLOPs for full ResNet-18 training, a $\sim 66.6\%$ reduction. Moreover, while baselines incur $\sim 5.4$ GFLOPs/sample in every epoch, FedAGE requires one frozen encoder forward pass (1.8 GFLOPs/sample) plus classifier head training ($5.7 \times 10^{-5}$ GFLOPs/sample) in the first epoch, and only classifier head training thereafter, as illustrated in Fig. 2(c). Over multiple epochs, the per-sample cost of FedAGE converges to nearly zero, making it highly efficient for resource-constrained edge environments.

## C Privacy Analysis of FedAGE

The architectural design of FedAGE, which relies on sharing latent embeddings from a frozen encoder in a single communication round, provides significant inherent privacy advantages over traditional federated learning frameworks that iteratively exchange gradients. This section analyzes the privacy properties of FedAGE from an information-theoretic perspective and evaluates its resilience against established attack vectors.

### C.1 Information-Theoretic Privacy Foundations

The core privacy benefit of FedAGE is grounded in the **Information Bottleneck (IB) principle** (Tishby et al., 2000) (Tishby & Zaslavsky, 2015) (Makhdoumi et al., 2014). The goal of the IB framework is to learn a compressed representation, $z$, of an input variable, $x$, that is maximally informative about a target variable, $y$, while being minimally informative about $x$ itself. This trade-off is formalized by optimizing the IB Lagrangian:

$$\mathcal{L}_{\text{IB}} = I(z; y) - \delta I(x; z) \tag{7}$$

where $z = E(x)$ is the latent embedding, $y$ is the class label, and $\delta$ is a parameter controlling the compression-utility trade-off.

In FedAGE, the frozen pre-trained encoder, $E(\cdot)$, acts as a fixed information bottleneck. Its purpose is to create a representation that preserves task-relevant features (maximizing $I(z; y)$) while compressing the input and discarding irrelevant information (minimizing $I(x; z)$). Privacy inference attacks, such as membership and attribute inference (Shokri et al., 2017) (Yan et al., 2025), succeed precisely by exploiting this extraneous information about the input $x$ that is leaked into a model's outputs or shared parameters. By design, our framework constrains this information channel, thereby inherently limiting the information available to an adversary.

### C.2 Architectural Privacy by Design

FedAGE's architecture translates this information-theoretic principle into practical privacy defenses.

**Frozen Encoder as a Privacy Filter.** The use of a public, pre-trained encoder is a critical design choice in the proposed FedAGE. Since the encoder's parameters are not trained on any client's private data, they cannot memorize client-specific features. The encoder functions as a deterministic, non-linear transformation that maps high-dimensional private data into a lower-dimensional latent space. This process has two key effects:

1. *Information Obfuscation:* The encoder's many-to-one mapping ensures that multiple distinct inputs can result in similar latent representations, creating ambiguity that complicates an attacker's ability to uniquely identify or reconstruct a specific input.

2. *Dimensionality Reduction:* Projecting an input data to a lower-dimensional space inherently discards information, making the inverse problem of reconstructing the original high-dimensional data from the embedding computationally challenging and often ill-posed.

**Mitigation of Gradient-Based Attacks.** Traditional FL is vulnerable to **gradient inversion attacks** (Zhu et al., 2019), where an adversary can reconstruct training data with high fidelity from shared gradients. FedAGE is intrinsically resistant to this entire class of attacks because it does not share gradients related to the frozen encoder. An attacker would need to invert the fixed, complex, and non-linear encoder function without access to its gradients, a significantly more difficult (Fredrikson et al., 2015).

**Resilience to Membership Inference Attacks (MIA).** MIAs typically exploit the fact that models exhibit higher confidence on data they were trained on ("members") versus unseen data ("non-members") (Shokri et al., 2017). In FedAGE, the server only receives latent embeddings and the weights of the small classifier head. It has no direct access to the client's model's confidence scores on its local data, disrupting the primary information channel used by standard black-box MIAs. Furthermore, the single-round communication protocol prevents attackers from observing the evolution of model updates over time, a technique used in more advanced attacks to amplify leakage signals.

### C.3 EMPIRICAL RESILIENCE TO BASIC MODEL INVERSION ATTACKS

To empirically validate the theoretical privacy claims, we subject our framework to a white-box model inversion attack (Fredrikson et al., 2015). In this threat model, we assume a worst-case adversary who has intercepted a client's latent embedding, $z$, and also possesses complete knowledge of the public, frozen encoder architecture $E(\cdot)$. The adversary's goal is to reconstruct the original private image, $x$, from this information.

The attack is formulated as an optimization problem, where the adversary attempts to find an image, $x'$, from target embedding $z$ that minimizes the distance between its embedding and the target embedding. We implement this by initializing a random noise image and iteratively updating its pixels to minimize a loss function $\mathcal{L}_{\text{TV}}$ composed of the mean squared error (MSE) between the embeddings and a total variation (TV) regularizer (Rudin et al., 1992), which encourages spatial smoothness, as described in. The objective is formally:

$$x' = \arg\min_{x'} \left( \|E(x') - z\|_2^2 + \lambda \cdot \mathcal{L}_{\text{TV}}(x') \right), \tag{8}$$

where $\lambda$ is a hyperparameter balancing the two terms. For our experiments, we set $\lambda = 10^{-4}$. As demonstrated in Figure 5, the attack consistently fails to reconstruct any meaningful information. The output is indistinguishable from random noise, revealing nothing about the semantic content of the original image. This failure is a direct consequence of the severe, lossy compression performed by the encoder. The mapping from the high-dimensional pixel space (e.g., $3 \times 32 \times 32$) to a low-dimensional latent space is a many-to-one function. An infinite number of potential inputs, including structured images and random noise, can be mapped to nearly identical points in the latent space. The optimization process successfully finds an input that satisfies the loss objective, but which bears no resemblance to the original data. This result provides strong empirical evidence that the encoder acts as a robust information bottleneck, effectively thwarting reconstruction and validating the privacy-by-design principles of our method.

### C.4 EXTENSION FOR FORMAL PRIVACY GUARANTEES AND THE UTILITY TRADE-OFF

The inherent privacy of FedAGE stems from its architecture, which presents computational and information-theoretic hurdles to an adversary. For applications requiring formal and provable guarantees, the framework can be extended by integrating mechanisms like *Differential Privacy (DP)* (Dwork et al., 2006) (Dwork & Roth, 2014).

A direct method to achieve this is by applying *local differential privacy*, where clients inject calibrated noise into their latent embeddings before transmission. This extension ensures that the shared data does not unduly reveal information about any single client's data point, providing a rigorous defence against a powerful adversary. However, this formal guarantee introduces the well-established *privacy-*

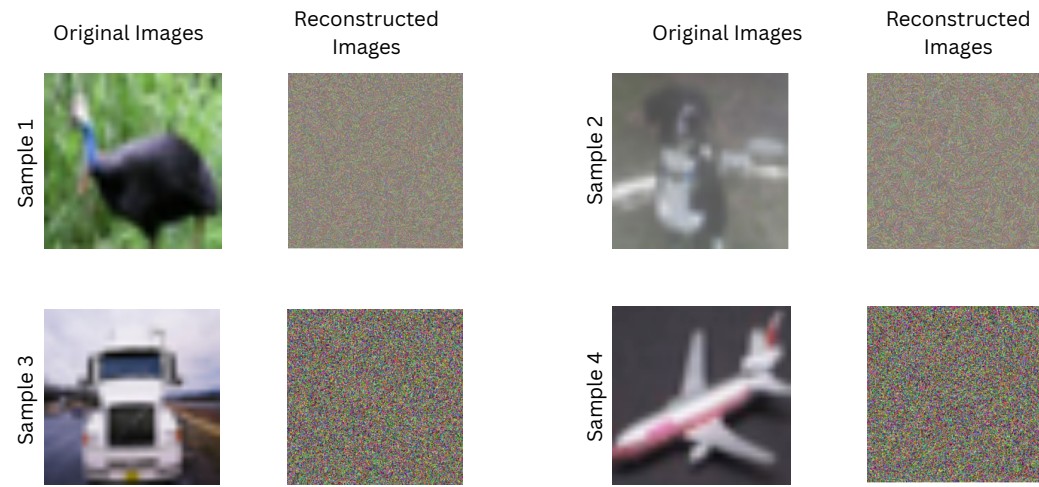

Figure 5: Visualisation of original images VS reconstructed images

*utility trade-off*: stronger privacy protection (achieved with a smaller privacy budget, $\epsilon$) requires more noise, which can in turn degrade the utility, i.e., the accuracy of the final global model (Bagdasaryan et al., 2019).

We empirically validate this trade-off using various privacy budgets for the CIFAR-10 dataset, with n=10 clients and $\alpha = 0.01$, while keeping all other hyperparameters constant. The results, presented in Figure 6, clearly illustrate the expected relationship. Under the most stringent privacy guarantees ($\epsilon \leq 1.0$), the model's utility is significantly constrained by the addition of calibrated noise. As the privacy budget is relaxed (i.e., $\epsilon$ increases), we observe a monotonic improvement in accuracy. This trend culminates in the no DP baseline ($\epsilon = \infty$), which achieves the highest accuracy.

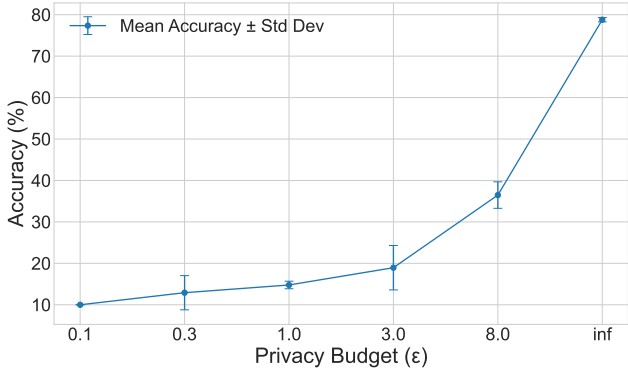

Figure 6: Plot showing the relationship between DP epsilon ($\epsilon$) and model accuracy.

## D   LLM USAGE

We acknowledge the use of large language models (LLMs), such as ChatGPT and Gemini, for assistance in paraphrasing and improving the grammatical clarity of the manuscript. These tools were not involved in the scientific contributions, experimental design, analysis, or validation of this work.

