# OpenReview forum: "One-Shot Federated Aggregation of Generalized Embeddings for Edge Environments"
_ICLR.cc/2026/Conference — ICLR 2026 Conference Withdrawn Submission_

### Official Review · Reviewer_GnA6 · 2025-10-28

**Soundness:** 2
**Presentation:** 3
**Contribution:** 2
**Rating:** 2
**Confidence:** 4

**Summary:**

This paper proposes FedAGE, a one-shot federated learning framework that mitigates the heavy communication and computation costs of traditional multi-round federated learning. In FedAGE, each client uses a shared frozen encoder to generate latent embeddings and trains only a lightweight classifier. Clients share these embeddings and classifiers with the server once, which then performs progressive and ensemble knowledge distillation to aggregate global knowledge efficiently. Experiments on five benchmark datasets demonstrate that FedAGE achieves higher accuracy and reduces client computation compared to OFL baselines.

**Strengths:**

1.	The idea of combining a frozen shared encoder with progressive and ensemble distillation in a one-shot setting is well-motivated.
2.	The paper effectively addresses both communication and client-computation bottlenecks in FL.
3.	The paper is well-organized and easy to follow.

**Weaknesses:**

1.	The method’s success highly depends on the quality and domain alignment of the pretrained encoder. The paper acknowledges this limitation but lacks quantitative analysis (e.g., results when encoder and local data distributions differ strongly).
2.	Although client computation is reduced, the server must handle all latent embeddings and execute sequential distillation, which could become a scalability bottleneck. No analysis is given on memory or runtime at the server.
3.	While the focus is on one-shot FL, it would strengthen the paper to include a lightweight multi-round FL baseline to contextualize the trade-off between communication efficiency and accuracy.
4.	Privacy analysis of this paper is relatively simple. The proposed method requires all clients to share their representations, which may incur new privacy issues. Despite some traditional attack methods being applied to illustrate the privacy performance of the proposed method, more powerful and fresh attack methods should be included to improve the convincing.
5.	While the authors claim that the proposed method can “mitigate catastrophic forgetting”, the paper lacks dedicated ablation studies or quantitative evidence to substantiate this claim, leaving its actual effectiveness unclear.

**Questions:**

Please see the weaknesses.

---

### Official Review · Reviewer_jtvH · 2025-10-30

**Soundness:** 3
**Presentation:** 3
**Contribution:** 3
**Rating:** 6
**Confidence:** 4

**Summary:**

FedAGE introduces a one-shot federated learning framework that shifts most of the computational burden to the server. Clients use a shared frozen pretrained encoder to generate latent embeddings and only train a lightweight classifier head.  At the server, progressive knowledge distillation sequentially aggregates knowledge from all clients to form a global model. The results from doing an evaluation on 5 datasets show significant improvement over existing baselines and can handle high heterogeneous partitioning. Its design is also computationally less expensive for clients, which makes it ideal for edge devices.

**Strengths:**

Only the classifier head is trained, drastically reducing trainable parameters, which is beneficial for resource-constrained edge devices.

An effective distillation technique on the server.

Maintains stable accuracy across varying degrees of data heterogeneity and datasets.

Works with multiple backbone architectures (ResNet-18, MobileNet, SqueezeNet)

**Weaknesses:**

FedAGE’s low client-side computation largely relies on a frozen pretrained encoder. Thus, its efficiency depends on the availability of suitable pretrained models. In domains without such models, e.g., sensor or general-purpose audio data common in edge FL [1,2], clients would need to train the encoder themselves, significantly increasing computation and reducing the advantages for resource-constrained devices, which limits FedAGE’s applicability.

The scalability results with increasing clients are only shown for the CIFAR-10 dataset. How do your results vary across datasets for an increasing number of clients with increasing data heterogeneity?

The sequential progressive distillation shifts heavy computation and memory load to the server, potentially limiting scalability and making the approach less practical for large-scale FL deployments - a large number of clients.

While the paper compares FedAGE to several recent OFL baselines, it misses some important recent works that are relevant to efficient one-shot FL. For instance, IntactOFL [3] and FedTMOS [4] also reduce client-side computation.

[1] Alexander Brecko, Erik Kajati, Jiri Koziorek, and Iveta Zolotova. Federated learning for edge computing: A survey. Applied Sciences, 12(18), 2022. ISSN 2076-3417. doi: 10.3390/app12189124.

[2] Berrenur Saylam and Özlem Durmaz İncel. Federated learning on edge sensing devices: A review, 2023.

[3] Zeng Hui, Xu Minrui, Zhou Tongqing, Wu Xinyi, Kang Jiawen, Cai Zhiping, and Niyato Dusit. One-shot-but-not-degraded federated learning. In Proceedings of ACM International Conference on Multimedia (ACMMM), 2024.

[4] Shannon How Shi Qi, Jagmohan Chauhan, Geoff V. Merrett, and Jonathon Hare. FedTMOS: Efficient one-shot federated learning with tsetlin machine. In The Thirteenth International Conference on Learning Representations, 2025

**Questions:**

FedAGE uses progressive knowledge distillation at the server, sequentially distilling each client into the student model. Could the authors clarify the trade-offs compared to standard knowledge distillation? In particular, how does progressive KD affect accuracy and server computation relative to aggregating all client soft labels at once?

Does the order of clients during the progressive KD affect the final model performance? If yes, it makes sense to show the results of the FedAGE robustness to the ordering.

Since FedAGE relies on a frozen pretrained encoder, how would the method perform if a non-pretrained encoder were used? How does this impact client-side computation and performance?

For the baselines, did the authors use a pretrained ResNet-18 or train from scratch?

Which three seed values did you try and why?

---

### Official Review · Reviewer_g7zp · 2025-11-01

**Soundness:** 2
**Presentation:** 3
**Contribution:** 2
**Rating:** 2
**Confidence:** 4

**Summary:**

This paper proposes FedAGE, a one-shot federated learning (OFL) framework specifically designed for resource-constrained edge environments. Rather than requiring clients to transmit full model weights for aggregation, FedAGE has each client share only latent embeddings derived from a shared, frozen encoder, along with lightweight classifier parameters. The server then performs a progressive distillation process incorporating weighted soft labels, ensemble distillation, and a knowledge mixing strategy to build a global model from these embeddings and classifiers. The authors conduct extensive experiments across five standard benchmarks, analysing the impact of data heterogeneity, number of clients, and computational efficiency, and report improvements over several OFL baselines.

**Strengths:**

1. Aggregation Paradigm: By shifting client-server communication from model weights to frozen-encoder-derived embeddings and classifier heads, FedAGE reduces the computation burden on client devices. This approach is a good response to the practical constraints of edge computing.
2. Computational Efficiency: The reduction in client-side trainable parameters and computational overhead is substantial, with FedAGE requiring only a lightweight classifier instead of a full deep model, leading to significant savings in both parameters and FLOPs per sample.
3. Robustness to Non-IIDness and Client Scaling: FedAGE demonstrates consistent and strong performance across varying degrees of data heterogeneity and a broad range of client counts, maintaining clear advantages over alternative approaches.

**Weaknesses:**

1. Server-side Bottlenecks: While client efficiency is the core strength of FedAGE, the paper provides limited discussion on the computational and storage demands at the server end. Since the server is responsible for storing all client embeddings and performing sequential as well as ensemble knowledge distillation, it may become a scalability bottleneck as the number of clients or the size of embeddings increases. A quantitative analysis of the server’s memory, compute, and bandwidth requirements is needed.
2. Heavy reliance on a frozen encoder: The main motivation of FL is to train from private clients' datasets. Particularly, the presence of a suitable frozen encoder may be unrealizable in practice. Furthermore, this may introduce clear bias risks (pre-trained model not representative), and creates an attack surface (adversarial or poisoned pre-trained model).
3. Potential Overfitting to Benchmarks: The evaluation focuses primarily on standard classification datasets, with no experiments on real-world edge or cross-modal tasks such as audio, NLP, time series, or medical imaging. This narrow scope may restrict the perceived generalizability of FedAGE. Including results on more complex datasets, such as TinyImageNet, would provide stronger evidence of its robustness and applicability.
4. Insufficient Details on Encoder and Latent Representation: The paper would benefit from a more detailed description of the encoder architecture, such as the layer configurations, filter dimensions, and the final size of the latent representation. These details are essential for accurately assessing the claimed savings in communication and client-side training computation.
5. The idea is primarily a sub-idea of FedGKT [1], which has not been compared with. There are many works that are follow ups of FedGKT, the authors need to cover some of the recent ones in their comparisons.
6. The embeddings can leak data privacy. It has to be experimentally demonstrated how the proposed scheme is robust to model inversion and other attacks.
7. While not directly FL, there is another recent work that seeks to train small models at clients with support from server by offloading some intermediate activations with guaranteed DP privacy [2]. It seems the current paper is an extension of [2], as the final outputs from the fully connected layers are also shared with the server.
[1] Group Knowledge Transfer: Federated Learning of Large CNNs at the Edge (NIPS 2020). [2] All Rivers Run to the Sea: Private Learning with Asymmetric Flows (CVPR 2024).

**Questions:**

Please address the weaknesses carefully.

---

### Official Review · Reviewer_YLzm · 2025-11-03

**Soundness:** 3
**Presentation:** 3
**Contribution:** 3
**Rating:** 4
**Confidence:** 3

**Summary:**

This paper presents a one-shot federated learning approach, based on knowledge distillation. Instead of uploading gradients from clients to a server, this paper proposed to upload embeddings and classifiers, for better performance in OFL. Overall, this is a good work, showing better results against several approaches on public datasets. Furthermore, privacy issue is also discussed.  However, I have several concerns and questions on the proposed method. The main concern is that the application scenarios are very narrow, which relies on a strong pre-trained encoder. When the clients' data are very different to the distribution of pretrained model, this method is hard to apply. I vote this paper to "marginally below the acceptance threshold. But would not mind if paper is accepted".

**Strengths:**

1. Overall, this paper is organized well and presents stronger experimental results than previous work, including FedKD, FedCVAE, Co-Boosting.

2. The motivation is clear. Even though previous one-shot federated learning approaches can reduce the communication costs, however, the performance is hard to guarantee.

**Weaknesses:**

1. In Alg 1, the name of  variables are confusing. For example, in line 8, what is P_{teacher}^{(i)}  and what is P_{s}^{(i-1)}. Those variables are not mentioned. Besides, P_{I} is defined earlier, which is different to P_s^{(i-1)}.

2. In order to reduce the communication cost, the classifier needs to be small. The most information is encoded in the encoder. When we need to fine-tune the encoder, this method has the risk of failure of fine-tuning the encoder.

3. Why is sequential distillation applied? Why not mixing all the clients and performing distillation? The server can have the embedding/classifier at the same time from each client. At least, this could be a good study (might be upper bound) to show the effectiveness of the sequential distillation.

4. Reducing communication costs is the main motivation of one-shot federated learning. However, communication cost is not quantitatively compared between different methods.

**Questions:**

The proposed method replies on a strong encoder. If the training data for the encoder is very different to the data on each client, how does this method work? Furthermore, this framework does not support training from scratch.

---

### Note · Authors · 2025-11-17

I have read and agree with the venue's withdrawal policy on behalf of myself and my co-authors.